# TSPAN8 as a Novel Emerging Therapeutic Target in Cancer for Monoclonal Antibody Therapy

**DOI:** 10.3390/biom10030388

**Published:** 2020-03-03

**Authors:** Kyun Heo, Sukmook Lee

**Affiliations:** Biopharmaceutical Chemistry Major, School of Applied Chemistry, Kookmin University, Seoul 02707, Korea; kyunheo@kookmin.ac.kr

**Keywords:** tetraspanin 8, prognostic marker, therapeutic target, cancer

## Abstract

Tetraspanin 8 (TSPAN8) is a member of the tetraspanin superfamily that forms TSPAN8-mediated protein complexes by interacting with themselves and other various cellular signaling molecules. These protein complexes help build tetraspanin-enriched microdomains (TEMs) that efficiently mediate intracellular signal transduction. In physiological conditions, TSPAN8 plays a vital role in the regulation of biological functions, including leukocyte trafficking, angiogenesis and wound repair. Recently, reports have increasingly shown the functional role and clinical relevance of TSPAN8 overexpression in the progression and metastasis of several cancers. In this review, we will highlight the physiological and pathophysiological roles of TSPAN8 in normal and cancer cells. Additionally, we will cover the current status of monoclonal antibodies specifically targeting TSPAN8 and the importance of TSPAN8 as an emerging therapeutic target in cancers for monoclonal antibody therapy.

## 1. Introduction

Monoclonal antibody therapy is one of the most effective therapeutic regimens against cancer. Since the Food and Drug Administration’s (FDA’s) approval of Muromonab-CD3 (OKT3), a mouse monoclonal antibody targeting CD3, recombinant DNA technology has led to the remarkable development of therapeutic antibodies, including humanized and fully human antibodies. Currently, numerous therapeutic antibodies are being widely used in clinics for treating hematological and solid tumors [1]. However, despite the rapid growth of the therapeutic antibody market during the last two decades, there remains only a limited number of available disease-specific biomarkers and therapeutic targets. Consequently, there is a substantial medical need for all-encompassing, effective cancer therapy. Nevertheless, molecular targets that can be used for therapy of certain cancer types do exist. For example, trastuzumab (Herceptin^®^), a humanized antibody targeting human epidermal growth factor receptor 2 (HER2), is being used for treating patients with HER2-positive breast cancer. Cetuximab (Erbitux^®^), a mouse/human chimeric antibody targeting anti-epidermal growth factor receptor (EGFR), received FDA approval for treating wild-type KRAS and EGFR-positive colorectal cancers [2]. A previous report has suggested trastuzumab to be effective for treating patients with HER2-positive breast cancers, representing approximately 30% of the total number of patients with metastatic breast cancers [3]. Conversely, approximately 20–30% of the total number of patients with metastatic colorectal cancers have been reported to be responsive to cetuximab [4]. Tumor progression and metastasis are complex and multistep processes. For example, in hematogenous metastasis, cancer cells first detach from the primary tumor; then migrate through the extracellular matrix, penetrate the basal membrane, lymphatic walls, and the endothelia of the vessel wall; and subsequently allow the cancer cells to circulate through the bloodstream. Circulating cancer cells can undergo hematogenous spread, settlement and growth in distant organs [5]. In this regard, the identification of a novel biomarker and therapeutic target, as well as understanding the molecular mechanism of metastatic cancer cells, are essential for overcoming the unmet medical need in current cancer therapy.

## 2. Clinical Relevance of Tetraspanins in Cancer

Tetraspanins are a family of proteins that consist of four transmembrane domains. They interact with themselves and other various cellular signaling molecules. These interactions result in the building of tetraspanin-enriched microdomains that efficiently mediate intracellular signal transduction. Since 1990, tetraspanins have been identified in humans, plants, fungi, *Drosophila melanogaster* (*D. melanogaster*) and *Caenorhabditis elegans*, (*C. elegans*), etc. [6]. In humans and rodents, 33 members of tetraspanin, referred to as TSPAN1-33, have been identified and are widely distributed throughout the cells and tissues [7]. 

A considerable number of studies have demonstrated that in normal physiological conditions, tetraspanins are closely implicated in the regulation of diverse biological processes such as cell adhesion, motility, proliferation, differentiation and immune cell functions [8,9]. In particular, in tumors, increasing evidence has suggested that tetraspanins interact with numerous binding partners to promote the progression and metastasis of a certain type of cancer cell. For example, TSPAN24 (CD151) was found to influence the association of integrin α6β4 with protein kinase C; activate STAT3 (a transcription factor constitutively active in many human malignancies); and support de novo tumor initiation, promotion and progression of skin squamous cell carcinoma [10]. The differential association of TSPAN29 (CD9) with CD315, CD316, epithelial cell adhesion molecule (EPCAM), claudin 1 and heparin-binding EGF-like growth factor (HB-EGF) can be related to the different effects on tumor cell invasion and metastasis [11].

Several tetraspanins, including tetraspanin 8 (TSPAN8), are known to be overexpressed in various types of cancers and have been associated with a poor prognosis, whereas some tetraspanins are downregulated (e.g., TSPAN5 in gastric cancer and TSPAN30 in lung cancer) and inversely correlated with overall patient survival [12,13]. The expression levels of some tetraspanins and their relationship with patient survival are cancer-type specific. For example, lower levels of TSPAN29 are present in several types of cancer, such as lung, breast, colon and pancreatic cancer, which display a positive correlation with good patient prognosis [14,15,16,17]. However, the expression pattern of TSPAN29 and overall patient survival are inversely correlated in gastric cancer [18]. Specifically, the level of TSPAN27 indicates a biphasic event in colon cancer development, consisting of upregulation during the early stages of colon cancer and complete loss of function once the tetraspanins are in metastatic tumor cells [19]. We have summarized the expression patterns and clinical relevance of tetraspanins in Table 1.

## 3. The Structure and Physiological Role of TSPAN8 in Cells

TSPAN8 is a member of the tetraspanin superfamily. It consists of four transmembrane domains (TM1-4); two extracellular loops, such as a small extracellular loop (SEL) between TM1 and TM2, and a large extracellular loop (LEL) between TM3 and TM4. It also contains three cytosolic domains including a short amino- and carboxy-terminal tail with a small intracellular loop between TM2 and TM3. Furthermore, LEL contains six conserved cysteine residues forming disulfide bonds that are essential for LEL conformation [60,61].

To date, different research groups have generated two monoclonal antibodies that target TSPAN8 as an antigen. Szala et al. were the first to report that TSPAN8 (CO-029) is a monoclonal antibody-defined cell surface glycoprotein of 27-34 kDa. They performed bio-panning with a monoclonal antibody to identify its target antigen from COS cells transfected with a cDNA library from the SW948 colorectal carcinoma cell line. They also showed that TSPAN8 is a tumor-associated antigen expressed in various cancer cell lines, including colorectal carcinoma, lung carcinoma, astrocytoma and ovarian carcinoma [62]. Later, Claas also reported that D6.1A recognized by a monoclonal antibody D6.1 is a rat homolog of CO-029 and plays a critical role in increasing the metastatic potential [63].

Several TSPAN8 expression profiling and knockout models have shown the physiological roles of TSPAN8 in cells. Champy et al. reported that human and mouse *TSPAN8* mRNA was highly expressed in cells in the digestive system, such as the stomach, small intestines and colon, where the genetic ablation of *TSPAN8* resulted in the reduction (15.6%) of body weight in only male mice. Although the detailed molecular mechanism is still unknown, they suggested that *TSPAN8* is at least in part associated with regulating body weight [64]. Furthermore, Zhao et al. reported that TSPAN8 is strongly expressed in the gastrointestinal tract, including the esophagus, stomach, small and large intestines, and some reproductive organs such as the ovary and testis, whereas *TSPAN8* knockout does not show any changes in organ structures and pathological phenotype. Additionally, Zhao et al. revealed evidence suggesting the physiological roles of TSPAN8. Firstly, they observed that *TSPAN8* knockout does not affect the immune response of leukocytes in response to mitogenic and antigenic stimuli compared to wild-type leukocytes, whereas it showed reduced leukocyte trafficking. Secondly, they showed a delayed and impaired vessel sprouting from the aortic rings in *TSPAN8* knockout mice. Furthermore, compared to wild-type endothelial cells, the migration of *TSPAN8* knockout endothelial cells was sharply reduced. Lastly, delayed wound healing was also observed in *TSPAN8* knockout mice. In summary, these results suggest that TSPAN8 may be necessary for the regulation of leukocyte trafficking, angiogenesis and wound repair [65] (Figure 1).

## 4. The Role of TSPAN8 in Cancer Progression and Metastasis

In the past 20 years, TSPAN8 appears to have played pivotal roles in the initiation and progression of multiple cancers. TSPAN8 is highly expressed in various cancerous tissues and mediates the proliferation, survival, invasion and metastasis of cancer cells (Figure 1). In this review, we examined the current status regarding the involvement of TSPAN8 in the development of several cancer types.

### 4.1. Pancreatic Cancer

In pancreatic cancer, both TSPAN8 and α6β4 integrin are highly expressed and correlate with increased tumor cell motility by promoting integrin activation through focal adhesion kinase, paxillin and Src recruitment [66,67]. Additionally, TSPAN8 is directly associated with CD9, CD81 and prostaglandin F2α receptor-regulatory protein (FPRP) in pancreatic carcinoma cells. This core complex is associated with α3β1 integrin, CD151, phosphatidylinositol 4-kinase (PI4K) and EPCAM at a higher level of integration [68]. Moreover, the overexpression of TSPAN8 in pancreatic cancer stimulates the upregulation of the expression of matrix metalloproteinases (MMPs), the angiogenic factor expression, as well as the secretion of urokinase-type plasminogen activator (Upa) and the expression of vascular endothelial growth factor (VEGF) and the VEGF receptor. All of these increases in expression comprehensively induces angiogenesis [69].

### 4.2. Colon Cancers

In colon carcinomas, TSPAN8 regulates colon cancer cell motility cooperation with the E-cadherin/p120-catenin (p120ctn) complex, which induces the selective recruitment of the α2β1 integrin pathways and interferes with small GTPase regulation [70]. TSPAN8 promotes the progression and metastasis of colorectal cancer by enhancing tumor cell movement and deregulating cell adhesions by altering the surface expression or activity of integrins and CD44 [71].

### 4.3. Gastric Cancers

The expression of TSPAN8 is increased both at the mRNA and protein levels in gastric cancer tissues and is associated with poor survival. The knockdown of *TSPAN8* attenuates the effects of the EGFR on gastric cancer cell proliferation and invasion [25,72]. TSPAN8 promotes gastric cancer cell growth and metastasis through the activation of the extracellular-signal-regulated kinase/mitogen activated protein (ERK/MAPK) pathway [73]. Moreover, TSPAN8 impairs the sensitivity of gastric cancer cells to chemotherapeutic agents by mediating Wnt/β-catenin activity [74].

### 4.4. Liver Cancers

TSPAN8 is frequently found and overexpressed at both the mRNA and protein levels in hepatocellular carcinoma (HCC) and TSPAN expression is correlated with intrahepatic spreading [75]. TSPAN8 could promote HCC migration and invasion in vitro by increasing the expression of ADAM12m, a membrane integrating disintegrin and metalloproteinase expression [26]. TSPAN8 also plays a critical role in mediating astrocyte-elevated gene-1 (AEG-1)-induced invasion and metastasis [76].

### 4.5. Lung Cancers

TSPAN8 is found to be overexpressed in non-small cell lung cancer (NSCLC) cells compared with normal human bronchial epithelial cells. The knockdown of TSPAN8 reduces cell viability and proliferation, while overexpression of TSPAN8 enhances cell viability and proliferation in NSCLC cells. TSPAN8 knockdown leads to G_1_ arrest and apoptosis by downregulating CDK2, CDK4 and Cyclin D1 and upregulating BCL2 Associated X, Apoptosis Regulator (Bax) and Poly (ADP-ribose) polymerase (PARP) [77].

### 4.6. Breast Cancers

TSPAN8 protein is present in the majority of human primary breast cancer lesions and metastases in different organs (the brain, bone, lung and liver). TSPAN8 enhances cell–cell adhesion, proliferation, mesenchymal–epithelial transition, radiation resistance and extracellular vesicle release [78].

### 4.7. Ovarian Cancers

TSPAN8 is overexpressed in epithelial ovarian cancer (EOC) tissues and correlates with poor survival. The knockdown of TSPAN8 or competition with purified TSPAN8-LEL protein inhibits EOC cell invasion. Moreover, the TSPAN8-blocking antibody significantly reduced the incidence of EOC metastasis in vivo [28].

### 4.8. Gliomas

The expression level of TSPAN8 is elevated in malignant glioma tissues and is correlated with tumor grading. The knockdown of TSPAN8 reduces proliferation and migration and increases the sensitivity of temozolomide (TMZ)-induced cell death and the apoptosis of glioma cells [79]. TSPAN8 forms a complex with Rapamycin-insensitive companion of mammalian target of rapamycin (RICTOR), a key component of mammalian target of rapamycin complex 2 (mTORC2), and integrin α3, which are required for mTORC2 activation and glioma cell migration [80]. Moreover, GSK621, a novel AMP-activated protein kinase (AMPK) activator, has shown anti-cancer activity in glioma cells by the degradation of TSPAN8 [81].

### 4.9. Melanomas

TSPAN8 is upregulated at the mRNA and protein levels in melanoma cells, where its levels are correlated with an invasive phenotype. The functional role of TSPAN8 is enhancing invasive outgrowth from tumor spheroids within matrigel without affecting cell proliferation or survival [82]. Lung cancer metastasis-related protein 1 (LCMR1) enhances the TSPAN8 expression and promotes the loss of melanoma cell-matrix adherence (mainly through β1-integrin-ILK axis) and increases in invasion in vitro and tumorigenicity in vivo [83,84]. p53 is a direct transcriptional repressor of TSPAN8 expression and regulates melanoma cell invasion in a TSPAN8-dependent manner. [85]. In contrast, β-catenin triggers the direct transcriptional activation of TSPAN8 expression, leading to melanoma invasion [86].

### 4.10. Esophageal Cancers

TSPAN8 is overexpressed in esophageal carcinomas and esophageal carcinoma cell lines with high-invasive potential. Exogenous expression of TSPAN8 promotes cell migration and invasion in esophageal carcinoma cell lines, as well as enhances the invasion and lung metastasis in esophageal tumor xenograft mouse models. ADAM12m is a key matrix degradation enzyme involved in TSPAN8-promoted invasion and metastasis [87].

### 4.11. Nasopharyngeal Cancers

TSPAN8 is overexpressed in nasopharyngeal carcinoma (NPC) tissues, where it is poorly differentiated and highly-metastatic. In NPC patients, the expression of TSPAN8 is associated with a reduced survival rate. The overexpression of TSPAN8 in NPC cells promotes cell migration, invasion and metastasis through the protein kinase B (Akt)/MAPK pathway [29].

### 4.12. Cancer Stem Cells

The expression level of TSPAN8 is upregulated in breast cancer stem cells. In these cells, TSPAN8 enhances Sonic Hedgehog (SHH) signaling, stabilizes the expression of protein patched homolog 1 (PTCH1) by recruiting ataxin-3 (ATXN3), and promotes cancer stemness [27].

## 5. TSPAN8 as an Emerging Therapeutic Target in Cancer for Antibody Therapy

The monoclonal antibody is a powerful tool for validating the role of a target protein as a potential therapeutic target for antibody therapy. To date, there is increasing evidence of the development of monoclonal antibodies which are specific to TSPAN8 and their roles in cancer treatment through a variety of in vitro and in vivo efficacy and toxicity evaluation studies (Figure 1).

### 5.1. Murine Monoclonal Antibody

Initially, TSPAN8 antibodies were generated as mouse monoclonal antibodies using hybridoma technology, which was developed by George Kohler and Cesar Milstein [88]. In 1979, Koprowski et al. first reported the generation of fifteen colorectal carcinoma-specific mouse antibodies [89]. A decade later, Szala et al. identified one monoclonal antibody that could specifically recognize CO-029 [62]. Greco et al. also generated TS29, a mouse monoclonal IgG1 antibody to CO-029, by immunizing human colon cancer cells such as Isreco3 and LoVo cells. In addition, they reported that this antibody showed a 75% motility reduction of CO-029-overexpressing human colon cancer cells [70]. Ailane et al. generated TS29.2, an IgG2b type of TS29, and found that with early treatment, this antibody inhibits the growth of CO-029-overexpressing tumor cells by up to 70% [90]. Stuhlmiiler et al. also generated a mouse monoclonal antibody (D6.1) by immunizing athymic nude mice with partially purified human melanoma tumor-associated antigens [91]. Subsequently, Claas et al. showed that this antibody is effective in modulating several tumor cell functions. For example, this antibody could reduce the adhesion of D6.1A-overexpressing rat adenocarcinoma cells to substrates such as plastic, hyaluronic acid and collagen type I, whereas it did not influence cell–cell adhesion. Moreover, the proliferation of D6.1A-overexpressing rat adenocarcinoma cells was inhibited by this antibody. Finally, through in vivo animal models, they confirmed that D6.1 could reduce the tumor growth as well as consumption coagulopathy of D6.1A-overexpressing rat adenocarcinoma cells [63]. Gesierich et al. also found that D6.1 inhibited D6.1A-induced angiogenesis [69].

### 5.2. Human Monoclonal Antibody

In a previous paper, we identified that TSPAN8-LEL is a critical domain for regulating metastatic colorectal cancer cell (mCRC) invasion using several biochemical analyses. First, we silenced *TSPAN8* via *TSPAN8* small interfering RNA (siRNA) into HCT116 mCRC cells and observed that *TSPAN8* knockdown reduced HCT116 cell invasion. Second, TSPAN8-overexpressing COS-7 cells showed an increased invasive nature. Third, a competition assay with purified Fc fusion proteins of each loop (TSPAN8-LEL-Fc and TSPAN8-SEL-Fc) revealed that HCT116 cell invasion was specifically blocked by TSPAN8-LEL-Fc, but not by TSPAN8-SEL-Fc. Then, using phage display technology, we isolated four fully human antibodies having different complementarity determining regions (CDRs) from a synthetic human antibody library. Among them, we selected one antibody having the strongest binding affinity to TSPAN8-LEL (K_D_ = 0.6 nM). We also revealed that this antibody specifically recognizes the amino acid residues 140–205 of TSPAN8-LEL in a conformation-dependent manner. Lastly, we found that the antibody inhibited the invasion of mCRC cells more potently than that of non-mCRC cells. In summary, these findings provide the first evidence that TSPAN8-LEL could be a potential therapeutic target in mCRC cell invasion [92].

In a different paper, we also reported that the high expression of TSPAN8 can be detected in approximately 52% of patients with EOC, and it is correlated with a reduced survival rate. We further revealed that TSPAN8-LEL is essential for EOC cell invasion through *TSPAN8* siRNA knockdown and a competition assay with purified TSPAN8-LEL-Fc. Moreover, we found that anti-TSPAN8 IgG not only specifically inhibited the invasion of EOC cell lines, including SNU8, SNU251 and SK-OV3 cells, but also wholly and dramatically blocked ovarian cancer cell metastasis by up to around 35% with a single dose. The mode of action of this antibody is likely to be the induction of potent internalization of cell surface TSPAN8 on EOCs. We observed the rapid internalization of the antibody and its co-localization with lysosome within 30 min. Furthermore, we found that antibody treatment can induce the specific down-regulation of cell surface TSPAN8 on EOCs, suggesting that our TSPAN8-specific antibody induced rapid internalization and concomitant down-regulation of cell surface TSPAN8 that is closely associated with EOC cell invasion and metastasis [28]. Conclusively, these observations indicate that TSPAN8 is not only a prognostic biomarker of EOCs, but also a therapeutic target for antibody therapy.

### 5.3. Radioisotope-Conjugate Antibody

Radioimmunotherapy is a promising therapeutic strategy for cancer therapy. To date, the FDA has approved two radioisotope-conjugated antibodies. The first is Ibritumomab tiuxetan (Zevalin^®^), an anti-CD20 monoclonal antibody radiolabeled with Yttrium-90, and the other is Iodine 131I-tositumomab (Bexxar^®^), an anti-CD20 monoclonal antibody radiolabeled with Iodine-131 [93]. Furthermore, a radioconjugate 131I-omburtamab, an anti-B7-H3 monoclonal antibody, is being evaluated in clinical studies [94]. Recently, Maisonial-Besset et al. were the first to generate the TSPAN8 monoclonal antibody (TS29.2) radiolabeled with indium-111 or lutetium-177. Through biodistribution studies, they found that these radiolabeled antibodies are highly specific to mCRC cells in nude mice bearing HT29 or SW480 colorectal carcinoma (CRC) xenografts. They also demonstrated that these antibodies are effective in specifically inhibiting tumor growth in HT29 tumor-bearing mice. In summary, these findings suggest that TSPAN8 is a promising therapeutic target for radioimmunotherapy in a broad range of TSPAN8-expressing cancers, including melanoma, glioma, hepatocellular carcinoma, ovarian and gastric cancers as well as aggressive CRC tumors [95].

## 6. Conclusions

To date, several reports have shown that TSPAN8 is a prognostic marker and plays a critical role in the progression and metastasis of cancer cells. Simultaneously, many murine and human monoclonal antibodies have been generated to evaluate the role of TSPAN8 in cancers for antibody therapy. In conclusion, currently available and accumulating evidence has led us to speculate that antibody-based targeting of TSPAN8 may be an effective strategy to suppress tumor progression and metastasis. Therefore, TSPAN8 can be considered a potential therapeutic target in TSPAN8-overexpressing cancers for antibody therapy. However, for future clinical trials, we need more careful consideration of safety in relation to on-target, off-tumor effects and variability in the patient population.

## Figures and Tables

**Figure 1 biomolecules-10-00388-f001:**
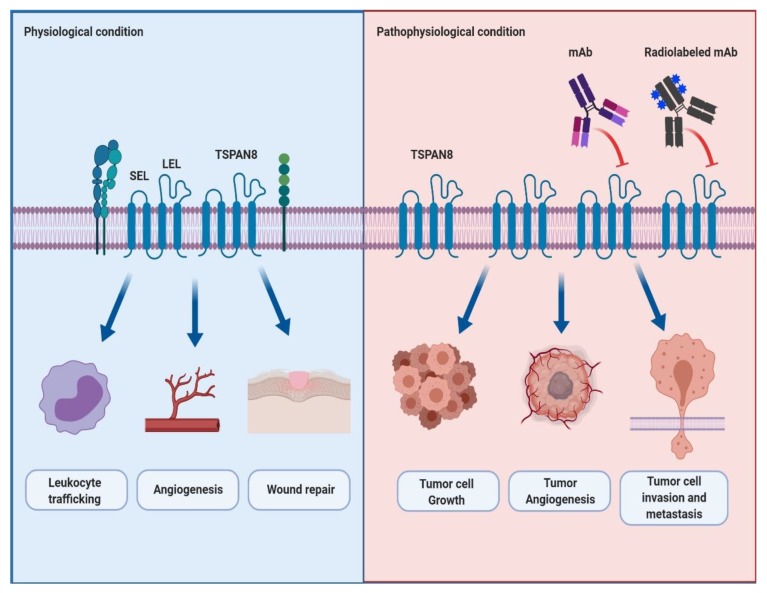
The schematic representation of the physiological and pathophysiological roles of TSPAN8 and the effect of a newly developed antibody targeting TSPAN8 in TSPAN8-mediated cancer progression and metastasis. Under physiological conditions, TSPAN8 interacts with itself and other binding proteins to efficiently convey outside signals to the inside of the cell. It plays a key role in the regulation of many cellular functions such as leukocyte trafficking, angiogenesis and wound repair. Under pathophysiological conditions such as cancers, it has been well-known that TSPAN8-overexpression is closely associated with the cell growth, angiogenesis, and invasion and metastasis of tumor cells. Furthermore, it is also believed that the monoclonal antibody or radiolabeled monoclonal antibody to TSPAN8 may be effective in suppressing TSPAN8-mediated tumor progression and metastasis. Abbreviation: mAb, monoclonal antibody.

**Table 1 biomolecules-10-00388-t001:** Correlation between tetraspanins and clinical prognosis in human cancers.

Tetraspanins	Cancer Types Correlated with Poor Prognosis	Cancer Types Correlated with Good Prognosis	References
TSPAN1 (NET-1)	GC, CRC, PCC		[20,21,22]
TSPAN2	Lung ADC		[23]
TSPAN5 (NET-4)		GC	[12]
TSPAN7 (Talla1, TM4SF2, CD231)	NSCLC		[24]
TSPAN8 (Co-029)	GC, HCC, BC, EOC, NPC		[25,26,27,28,29]
TSPAN9 (NET-5, PP1057)	GC		[30]
TSPAN13 (NET-6)		BC	[31]
TSPAN15 (NET-7)	ESCC		[32]
TSPAN17	GBM		[33]
TSPAN24 (CD151, Peta3)	HCC, GBC, BC, HGSC, NSCLC, PC, GC, CRC		[34,35,36,37,38,39,40]
TSPAN27 (CD82, KAI1)	CCRCC	ESCC, NSCLC, PCC, BC, GBC, CRC	[17,41,42,43,44,45,46]
TSPAN28 (CD81, Tapa1)	BC, MM, AML	GBC	[47,48,49,50]
TSPAN29 (CD9, p24)	GC	NSCLC, BC, CRC, PCC, NHL, ESCC, OSCC, EM, HNSCC, UBC, MM, GBC, NB, AML	[14,15,16,17,18,42,51,52,53,54,55,56,57,58,59]
TSPAN30 (CD63)		Lung ADC	[13]

Abbreviations: GC (gastric carcinoma), CRC (colorectal carcinoma), PCC (pancreatic carcinoma), Lung ADC (Lung adenocarcinoma), NSCLC (non-small-cell lung cancer), BC (breast cancer), EOC (epithelial ovarian cancer), NPC (nasopharyngeal carcinoma), ESCC (esophageal squamous cell carcinoma), GBM (glioblastoma), HCC (hepatocellular carcinoma), GBC (gallbladder carcinoma), HGSC (high-grade serous ovarian cancer), PC (prostate cancer), CCRCC (clear cell renal cell carcinoma), MM (multiple myeloma), AML (acute myeloid leukemia), NHL (non-Hodgkin lymphoma), OSCC (oral squamous cell carcinoma), EM (endometrial cancer), HNSCC (head and neck squamous cell carcinoma), UBC (urothelial bladder carcinoma) and NB (neuroblastoma).

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
