# Peer review of "TSPAN8 as a Novel Emerging Therapeutic Target in Cancer for Monoclonal Antibody Therapy"

_biomolecules, 2020, doi:10.3390/biom10030388_

Round 1

Reviewer 1 Report

The review manuscript by Heo and Lee describe the involvement of TSPAN8 in cancer and present this protein as a potential molecualr target for selective cancer treatment. In general the review is well written and contains sufficiently detailed information. The included Table and Figure facilitate understanding of presented topic.

What I miss in this review is the more detailed descrition of TSPAN family. The authors should include chapter about tetranspanins - what are these proteins doing in healthy cells? What happens with the activity of these proteins in cancer cells? Why TSPANs may facilitate carcinogenesis on the molecular scale? Similarly, more focus on cell biology of TSPAN8 should be inculded before the detailed description of expression changes in distinct cancer types. The incorporation of such data will facilitate understanding of this manuscript,which the potential readers will certainly benefit from.

Author Response

Response to reviewers

We appreciate the detailed and helpful comments from the reviewers. Below, we have provided point-by-point responses to thoroughly address the concerns of each reviewer. We also performed intensive proofreading by professional proofreading company. In the revised manuscript, all of modified words and sentences are in red.

â–ª Comments from Reviewer #1

The review manuscript by Heo and Lee describe the involvement of TSPAN8 in cancer and present this protein as a potential molecualr target for selective cancer treatment. In general the review is well written and contains sufficiently detailed information. The included Table and Figure facilitate understanding of presented topic.

--> We appreciate the reviewer’s positive comments.

What I miss in this review is the more detailed description of TSPAN family. The authors should include chapter about tetranspanins - what are these proteins doing in healthy cells? What happens with the activity of these proteins in cancer cells? Why TSPANs may facilitate carcinogenesis on the molecular scale?

--> As suggested, we added the physiological and pathophysiological roles and the molecular mechanism of tetraspanins in the revised manuscript (line 57-67)

Similarly, more focus on cell biology of TSPAN8 should be included before the detailed description of expression changes in distinct cancer types. The incorporation of such data will facilitate understanding of this manuscript, which the potential readers will certainly benefit from.

--> In the original manuscript, to emphasize the role of TSPAN8 as a novel potential therapeutic target in cancers, we are initially aiming at comparing the physiological role of TSPAN8 (section 3) and pathophysiological role of TSPAN8 in cancer progression and metastasis (section 4). Later, for a better understanding, we also described the possibility of TSPAN8 as an emerging therapeutic target in cancer for antibody therapy by enumerating the current status of TSPAN8 antibody development (section 5).

Reviewer 2 Report

The authors present a review on the possibility of monoclonal antibody therapy of cancer using tetraspanin molecule TSPAN8 as a target. They summarize the differential expression of members of tetraspanin family in healthy cancer state, with the special emphasis on TSPAN8. The opportunities of using monoclonal antibodies that deplete cell membranes of TSPAN8 via internalization or are deleterious towards TSPAN8-overexpressing cells as radioconjugates. The review is systematically written and a valuable read for audience interested in antibody therapy and especially emerging targets. The manuscript could be improved by considering following minor remarks:

Line 31: molecular targets that can be addressed for therapy of certain cancer types

Line 37: that represent instead of consist

Line 38: word order: …% of patients are responsive to cetuximab.

Line 42: subsequently allow

Line 63: consisting of instead of such as

Line 120: matrix metalloproteinases

Line 122-123: The upregulation of expression…

Lien 140: by increasing the expression of…

Line 232: TSPAN8-LEL

Line 234: of TSPAN8 can be detected in…

Line 240: is likely to be the induction of potent internalization…; maybe form only one out of the 2 sentences describing the induction of internalization of TSPAN8 because they are very similar.

Line 253: this year, a radioconjugate 131I-omburtamab was mentioned in “Antibodies to watch” editorial as an interesting therapeutic antibody undergoing investigation – maybe you would consider mentioning it in the section on radioconjugates.

Line 263: No reference to Figure 1 can be found in the text.

Line 273: In the conclusion section, I would propose including comments on other aspects of using anti-TSPAN8 agents for therapy, such as safety issues, the extent of expression on non-cancer cells that could influence the on-target, off-tumor effects, and variability in the patient population.

Author Response

Response to reviewers

We appreciate the detailed and helpful comments from the reviewers. Below, we have provided point-by-point responses to thoroughly address the concerns of each reviewer. We also performed intensive proofreading by professional proofreading company. In the revised manuscript, all of modified words and sentences are in red.

â–ª Comments from Reviewer #2

The authors present a review on the possibility of monoclonal antibody therapy of cancer using tetraspanin molecule TSPAN8 as a target. They summarize the differential expression of members of tetraspanin family in healthy cancer state, with the special emphasis on TSPAN8. The opportunities of using monoclonal antibodies that deplete cell membranes of TSPAN8 via internalization or are deleterious towards TSPAN8-overexpressing cells as radioconjugates. The review is systematically written and a valuable read for audience interested in antibody therapy and especially emerging targets. The manuscript could be improved by considering following minor remarks:

--> We appreciate the reviewer’s positive comments.

<Minor points>

Line 31: molecular targets that can be addressed for therapy of certain cancer types

--> As suggested, we revised the sentence (line 31-32)

Line 37: that represent instead of consist

--> As suggested, we revised the sentence (line 37)

Line 38: word order: …% of patients are responsive to cetuximab.

--> As suggested, we revised the sentence (line 38-39)

Line 42: subsequently allow

--> As suggested, we revised the sentence (line 43)

Line 63: consisting of instead of such as

--> As suggested, we revised the sentence (line 76)

Line 120: matrix metalloproteinases

--> As suggested, we revised the word as plural (line 134)

Line 122-123: The upregulation of expression…

--> As suggested, we revised the sentence (line 134)

Lien 140: by increasing the expression of…

--> As suggested, we revised the sentence (line 154)

Line 232: TSPAN8-LEL

--> We corrected the word in the revised manuscript (line 247)

Line 234: of TSPAN8 can be detected in…

--> We added the words in the revised manuscript (line 249)

Line 240: is likely to be the induction of potent internalization…; maybe form only one out of the 2 sentences describing the induction of internalization of TSPAN8 because they are very similar.

--> As suggested, we revised the sentence (line 255-6)

Line 253: this year, a radioconjugate 131I-omburtamab was mentioned in “Antibodies to watch” editorial as an interesting therapeutic antibody undergoing investigation – maybe you would consider mentioning it in the section on radioconjugates.

--> As suggested, we added the sentence in the revised manuscript (line 268-9)

Line 263: No reference to Figure 1 can be found in the text.

--> We properly added the reference of Figure 1 in the revised manuscript (line 121, 125, 214)

Line 273: In the conclusion section, I would propose including comments on other aspects of using anti-TSPAN8 agents for therapy, such as safety issues, the extent of expression on non-cancer cells that could influence the on-target, off-tumor effects, and variability in the patient population.

--> As suggested, we added some sentences in the conclusion section (line 296-8)